# ELF5: A Molecular Clock for Breast Aging and Cancer Susceptibility

**DOI:** 10.3390/cancers16020431

**Published:** 2024-01-19

**Authors:** Masaru Miyano, Mark A. LaBarge

**Affiliations:** 1Department of Population Sciences, Beckman Research Institute, City of Hope, Duarte, CA 91010, USA; 2Center for Cancer and Aging, Beckman Research Institute, City of Hope, Duarte, CA 91010, USA; 3Center for Cancer Biomarkers Research, University of Bergen, 5007 Bergen, Norway

**Keywords:** ELF5, mammary epithelia, aging, breast cancer, breast-specific biomarker, early detection biomarker

## Abstract

**Simple Summary:**

Breast cancer is common among the women in the U.S., especially in older women. Genetic mutations and aging are major risk factors. The breast undergoes changes during life stages, and a gene called ELF5 is crucial. ELF5 helps mammary glands develop during pregnancy. As we age, ELF5 decreases, possibly increasing cancer risk. In normal breast development, ELF5 guides cells for lactation. In cancer, ELF5 acts differently in various subtypes, affecting cell growth and hormone sensitivity. Aging changes, like reduced *ELF5* and increased DNA modifications, may contribute to cancer risk. *ELF5* can be a “biological clock” indicating breast age and risk. It could refine prevention trial groups and show treatment effects. ELF5’s role needs validation for practical use in breast cancer prevention and early detection.

**Abstract:**

Breast cancer is predominantly an age-related disease, with aging serving as the most significant risk factor, compounded by germline mutations in high-risk genes like *BRCA1*/*2*. Aging induces architectural changes in breast tissue, particularly affecting luminal epithelial cells by diminishing lineage-specific molecular profiles and adopting myoepithelial-like characteristics. ELF5 is an important transcription factor for both normal breast and breast cancer development. This review focuses on the role of ELF5 in normal breast development, its altered expression throughout aging, and its implications in cancer. It discusses the lineage-specific expression of ELF5, its regulatory mechanisms, and its potential as a biomarker for breast-specific biological age and cancer risk.

## 1. Introduction

Breast cancer (BC) stands as the most prevalent cancer among women worldwide. Regions such as Australia, New Zealand, Western and Northern Europe, and Northern America exhibit the highest incidence rates of BC, exceeding 80 cases per 100,000 females, which is notably higher than in other parts of the world [1]. In the U.S., BC ranks as the most common form of cancer among women [2]. More than 75% of women who were diagnosed for BC for the first time in their lives are over 50 years old [3]; indeed, aging is the greatest risk factor for BC. High-risk women who are carriers of inherited germline mutations in genes such as *BRCA1*, *BRCA2*, and *PALB2* face a potentially seven-fold higher risk than those who are at average risk [4]. Although factors like parity, obesity, smoking, and alcohol consumption have been identified as additional risk factors, their increased risk is smaller than that associated with inherited germline mutations and aging [5,6]. Thus, germline mutations and aging are the predominant risk factors for BC.

The breast differs from other tissues in the human body because it has specialized functions related to lactation and milk production. Unlike other tissues, the breast undergoes significant changes during different stages such as puberty [7], the menstrual cycle [8], pregnancy [9], involution [10], and menopause [11]. Breast tissues comprise various types of cells, including epithelial cells, fibroblast cells, adipocytes, endothelial cells, and immune cells, and structures. The terminal duct lobular unit (TDLU) is an important structure in the breast, responsible for milk production and serving as the main origin of many BC precursors and cancers [12]. TDLU consists of two different types of epithelia: luminal epithelial cells and myoepithelial cells. Luminal epithelial cells are surrounded by myoepithelial cells, both derived from the same stem/bipotent progenitors, though luminal epithelial cells and myoepithelial cells show distinct functions. While myoepithelial cells have a contractile function and work as a tumor suppressor, the role of luminal epithelial cells is producing milk. 

Aging alters breast tissue architecture, with increased adipose cell proportions [11], basement membrane disruption [13], and epithelial composition shifts [14]. Lineage fidelity loss in luminal epithelial cells is a hallmark of aging [14,15,16,17], marked by reduced luminal epithelial cell-specific transcriptomes and methylomes, and the gain of myoepithelial cell-like features [17,18]. Although loss of fidelity with age is observed in both lineages, it is most pronounced in luminal epithelial cells. This is significant, as luminal epithelial cells are implicated as the cell-of-origin for age-related luminal BC subtypes [19]. Understanding these age-related molecular and cellular changes is vital for linking aging biology with BC biology and elucidating how aging make breast tissue susceptible to BC.

ELF5, E74-like ETS transcription factor, was cloned from mouse and human lung cDNA in 1998 as a member of the ETS transcription factor family [20]. *ELF5* is localized to the human chromosome 11p13. The chromosome loci 11p13-15, including *ELF5* gene, frequently undergoes loss of heterozygosity in several types of carcinomas including those in the breast, kidney, and prostate [20]. Extensive studies have characterized ELF5 as a crucial factor in early embryonic development, breast development during pregnancy, and BC development. Notably, our group has highlighted that the downregulation of *ELF5* expression is a signature of aging in breast tissue, potentially contributing to increased susceptibility to BC. This review aims to succinctly summarize the diverse biological roles of ELF5 in normal breast development, BC, and the aging process.

## 2. ELF5 in Normal Breast Development

*ELF5* is differentially expressed in luminal epithelial cells, but not myoepithelial cells [15,16,21]. The lineage-specific expression in luminal epithelia decreases with age (Figure 1A). *ELF5* is expressed in both luminal progenitors and luminal cells in mice [22,23], and the expression of ELF5 and ERα are mutually exclusive in mouse mammary glands [23]. 

Studies employing single-cell RNA sequencing (scRNA-seq) on human breast tissues have elucidated that ELF5 is expressed in ER− luminal cells—specifically, the so-called luminal alveolar secretory precursor (LASP) cells [17,24]—but is absent in ER+ luminal hormone-sensing (LHS) cells. This finding corroborates the mutual exclusivity of ERα and ELF5 expression in human mammary glands. While it is established that ELF5 represses ERα expression in BC cell lines [25], the mechanisms underlying ELF5-mediated downregulation of ERα remain unexplored. Transcriptome profiling has revealed seven distinct subgroups of LASP cells, with the subgroup expressing high levels of ELF5 primarily localized within lobules [24]. Contrasting evidence, however, indicates that progenitors from mammary ducts express higher levels of KRT15+ ELF5 compared to those from terminal duct lobular units (TDLU) [26]. Given the known sensitivity of luminal epithelial cells to their microenvironment [16] and intrinsic epithelial plasticity [27,28], the classification of these cells could vary among individuals and across menstrual cycles.

The National Center for Biotechnology Information (NCBI) database lists four different *ELF5* isoforms. Isoform variation arises from either the use of a unique first exon linked to distinct promoters or the exclusion of exon 4, which encodes the PNT domain that mediates protein–protein interactions. However, all isoforms retain the ETS DNA-binding domain. Functional distinctions among these isoforms have not been documented. *ELF5* expression is observed in disease-free tissues of the breast, bladder, head and neck, kidney, lung, and prostate [29], with the breast tissue exhibiting one of the highest expressions of ELF5 across various tissues. Tissue-specific expression patterns are evident for the isoforms [29]. Isoform 2 (NM_001422) is predominantly expressed, while isoform 3 (NM_001243080) shows low expression levels in luminal epithelial cells [15]. Notably, the expression of both isoforms diminishes with age.

Following the initial discovery and cloning of *ELF5* cDNA, investigations into ELF5 functions revealed its regulatory influence on genes specific to epithelial cells [30] and whey acidic protein (WAP) genes [31], as demonstrated through in vitro promoter assays. The research group led by Christopher Ormandy has conducted extensive studies on normal breast and BC development, employing a mouse model system. Studies utilizing genetically engineered mice underscored ELF5’s heightened expression during pregnancy, highlighting its critical role in the development of mammary gland alveoli. Observations in *ELF5* heterozygous mice during gestation revealed an accumulation of CD61+ luminal progenitors that are double positive for K14/K8, suggesting that ELF5 is pivotal in directing the fate of CD61+ luminal progenitors towards mature alveolar cells. Furthermore, ELF5 is implicated in the production of milk by modulating the regulation of milk proteins during pregnancy [23,32].

DNA methylation is one of the key epigenetic modifications to regulate gene expression without affecting DNA sequences. In a mouse model, the DNA methylation level is decreased on the *ELF5* promoter region in luminal epithelial cells during pregnancy, leading to increased expression of *ELF5*. Whereas, the *ELF5* promoter is methylated in myoepithelial cells, and *ELF5* expression is silenced [33]. This lineage-specific DNA methylation-based *ELF5* regulation mechanism is active also in primary human mammary epithelial cells (HMEC) derived from breast tissues [15,16]. Indeed, primary myoepithelial cells cultured with the methylation inhibitor 5-aza-2′-deoxycytidine exhibited a doubling of *ELF5* expression [16]. During early embryonic development, *ELF5* expression undergoes tight regulation, involving DNA methylation of its promoter in a developmental stage-dependent manner [34]. These findings suggest that the ON-OFF regulation of *ELF5* expression, linked to the level of DNA methylation, is essential for preserving cell-specificity. 

Throughout pregnancy, mammary glands undergo significant structural changes in response to prolactin and progesterone. Lee et al. elucidated a mechanism wherein RankL acts as a paracrine mediator, inducing *ELF5* expression upon exposure to progesterone [35]. This induction plays a role in the differentiation of luminal secretory cells from luminal progenitor cells. Since ELF5 overexpression rescues prolactin receptor deficient phenotype in mice [21], *ELF5* is a part of prolactin/prolactin receptor signaling. Mouse studies demonstrated that prolactin receptor activates *ELF5* expression [36], but not vice versa [21]. Using primary *ELF5*-null mammary epithelial cell cultures showed that ELF5 directly regulates *Stat5* to regulate WAP proteins in response to prolactin signaling; however, genes related to differentiation were not altered, indicating ELF5 indirectly regulates those genes [37]. These studies have demonstrated that positive regulation of *ELF5* via progesterone and prolactin signaling is the crucial process for developing functional breasts during pregnancy. 

As we describe in more detail later in this review, women who are carriers of *BRCA1* show a decline of *ELF5* expression. It has been reported that *BRCA1* mutation carriers suffer from poor or no milk production following their first pregnancy; however, they can regain the ability to breastfeed after subsequent pregnancies [38]. Interestingly, *ELF5* knockout and heterozygote female mice failed to produce milk during their first gestation, but *ELF5* heterozygote females regain the milk production after second or third gestations [37]. Haploinsufficiency of ELF5 expression in mice resembles women who are *BRCA1* mutation carriers in terms of *ELF5* expression, suggesting potential overlapping ELF5 function between mice and humans in terms of regulation of milk production by ELF5. 

It is known that ELF5 serves as a gatekeeper, either strengthening the commitment to the trophoblast lineage or redirecting epiblast cells away from the pathway [34]. SATB1 was shown to interact with ELF5 through long-range chromatin looping in mouse trophoblast [39]. SATB1 recruits chromatin remodeling complexes to promote epigenetic modifications for gene regulation [40] and serves as a genome organizer, regulating numerous genes by tethering regulatory elements to SATB1 binding sites through the formation of chromatin loops [41,42]. This facilitates the proximity of distant regulatory elements to the genes they regulate. SATB1 organizes T cell differentiation [43], epidermis development [44], early postnatal cerebral cortical development [45], and embryonic stem cell differentiation [46]. *SATB1* differentially expressed in luminal epithelial cells, and the lineage-specific expression is decreased with age [18]. This raises a potential regulatory mechanism by ELF5 in luminal epithelial cells; ELF5 may interact with SATB1 to orchestrate the regulation of its target genes in the maintenance of luminal epithelial cells and differentiation from luminal progenitors. Therefore, age-associated declines of both *ELF5* and *SATB1* expression may lead to the loss of proper gene regulation during aging.

## 3. ELF5 in Breast Cancer

*ELF5* expression undergoes significant changes across various cancers when compared to normal tissues [29]. Cervical, colon, rectal, and uterine cancers show increased *ELF5* levels, driven mainly by isoform 2 and, to a lesser extent, isoform 3. Head and neck, lung, and prostate cancers exhibit decreased *ELF5* expression. *ELF5* expression is nearly suppressed in three kidney carcinoma subtypes [29].

ELF5 functions as a tumor suppressor in bladder [47], kidney [48], ovary [49], and prostate [50] cancers. In BC, *ELF5* expression patterns are subtype specific and it exhibits bifunctional roles of promoting or suppressing cancers in a subtype-dependent manner [25,51]. The expression of *ELF5*, predominantly isoform 2, is higher in basal-like BC and lower in luminal A/B and HER2+ BCs as compared to normal breast tissue (Figure 1B) [29]. Like in normal epithelia, the expression is negatively correlated with DNA methylation in the promoter proximal region, with decreased and increased DNA methylation on the promoter in basal-like and luminal A/B BCs, respectively [15] (Table 1). As described later, DNA methylation on the *ELF5* promoter is increased with age. In luminal subtype BCs, elevated DNA methylation states were observed, and intriguingly, these states were found to be independent from age. This discovery implies that the age-dependent DNA methylation at the *ELF5* locus might act as a preparatory factor for the development of luminal subtype BCs. However, once cancer has initiated, the connection between age and *ELF5* expression or methylation appears to diminish.

In luminal A BC-subtype cell lines MCF-7 and T47D, ELF5 expression inhibits proliferation, whereas ELF5 promotes cell proliferation in basal-like BC cell lines, suggesting that the role of ELF5 could be cancer subtype-dependent [25,51]. The promotion of cell proliferation by ELF5 seems to occur through the induction of Cyclin D1 (CCND1) via the binding of ELF5 to the CCND1 promoter. However, the regulatory process is interrupted when ELF5 is acetylated, leading to the inhibition of CCND1 expression. Consequently, the varied acetylation status of ELF5 may contribute to subtype-dependent outcomes in breast cancer proliferation influenced by ELF5 expression [57]. Kalyuga et al. reported that tamoxifen-resistant luminal cancer cell lines increased *ELF5* expression, and increased expression of *ELF5* in luminal cancer cell lines drives a basal-like signature that leads to the suppression of sensitivity to estrogen [25]. Induction of ELF5 leads to rewiring FOXA1 and ER transcriptional networks to drive estrogen insensitivity by changing their binding sites, regulation of ER complex, and direct interaction with the ER complex [56]. ELF5 also works as a suppressor of epithelial-mesenchymal transition (EMT) through direct targeting of *Snail2* to suppress its expression, thereby inhibiting epithelial-mesenchymal transition [52]. Notably, in triple negative BC, the loss of *ELF5* promotes metastasis through activating IFN-γ signaling [53]. However, there is a potential contradiction in that induction of ELF5 to luminal cancer using the MMTV-PyMT mouse model shows that ELF5 increases the size and number of lung metastases [55]. ELF5 may have dual roles to suppress or drive EMT in a context-dependent manner (Table 1).

*ELF5* expression in luminal epithelial progenitor cells implicates a complex interplay between cellular phenotype and breast cancer progression, with implications for tumor heterogeneity and treatment resistance. In normal breast tissue, ER− luminal epithelial cells progenitors exclusively express ELF5 [23]. scRNA-seq confirms that cells expressing *ELF5* show a distinct profile from luminal hormone-sensing cells [17,24]. The Cancer Genome Atlas data analysis reveals generally low *ELF5* expression in luminal A/B BCs, with exceptions of higher expression in some cases [15,25]. However, one cannot completely rule out contamination from normal luminal epithelial cells to tumor samples as a cause for this variability. Notably, ER+ luminal cancer cell lines like T47D and MCF-7 express ELF5 and exhibit proliferation [58,59]. As individuals age, the population of ER+ luminal epithelial cells in normal breast tissue increases, but these cells show minimal proliferation. This trend complicates the maintenance of ER+ luminal epithelial cells in primary human mammary epithelial cell cultures over successive passages. Research indicates that culturing with TGFβ receptor inhibitors (TGFβR2i) can sustain ER+ luminal epithelial cells across passages [60]. TGFβR2i treatment enables cKit^high^ luminal epithelial cells progenitors, which are ER−, to differentiate into ER+ luminal epithelial cells and induces *ELF5* expression in ER+ luminal epithelial cells, suggesting that *ELF5*-expressing progenitors, regardless of their ER status, can give rise to ER+ luminal epithelial cells. This finding aligns with studies that suggest luminal ER− cells can differentiate into various phenotypes, including ER+ invasive ductal carcinomas [61]. Such plasticity could explain the diversity in ELF5 expression among ER+ cancers, potentially depending on whether the cancer cells originate from ER+/ELF5− or ER−/ELF5+ luminal epithelial cells progenitors. Since endocrine-therapy-resistant cells tend to induce ELF5 expression, ER+ cancer cells deriving from ER− luminal epithelial cells progenitors may be predisposed to developing resistance.

## 4. ELF5 in Aging and Susceptibility to Breast Cancer

One of the prominent age-dependent molecular changes in luminal epithelia is decreased expression of the *ELF5* gene associated with increased DNA methylation on its promoter region [15,16]. Age-dependent downregulation of *ELF5* expression in the mammary gland was also reported in mice [62] and non-human primates [63], indicating that *ELF5* downregulation with age may be a conserved aspect of aging. The similarity of the accumulation of KRT14 and KRT8 or KRT19 double-positive progenitors during pregnancy in the conditional *ELF5* knockout mice and during aging in human breasts suggests that the loss of *ELF5* expression during aging might cause increased proportions of luminal progenitor cells.

Emerging evidence from transcriptional profiling points to the downregulation of ELF5 as a pivotal driver of the aging phenotype in luminal epithelial progenitor cells, with potential implications for increased breast cancer susceptibility. Transcriptional profiling of luminal epithelial progenitor (luminal epithelial cells) cells from disease-free breast tissue in young to middle-aged women carrying *BRCA1* or *BRCA2* mutations reveals an accelerated aging phenotype. The differentially expressed genes in these high-risk women share a 35.3% overlap with aging signature genes, including *ELF5*, which is also found in average-risk women older than 50 years [64]. The observation that breast tissue at risk for cancer exhibits an aging phenotype suggests that this pattern arises independently of the women’s specific genetic makeup or their age-related breast cancer risk. In luminal epithelial progenitor cells from middle-aged, high-risk women, ELF5 expression decreases like it does with aging, and its promoter shows increased DNA methylation. These changes occur even when comparing these cells to those from average-risk, age-matched controls [15]. There are two reasons why age-associated downregulation of *ELF5* may be important to establish aging phenotypes. First, age-associated downregulation of *ELF5* in luminal epithelial cells is associated with age-specific changes in ELF5-target genes [16], indicating that the decline of *ELF5* is not a downstream event but an upstream one. Second, age-associated differential DNA methylation regions (DMRs), particularly those exhibiting heightened DNA methylation with age in luminal epithelial cells, are associated with an enrichment of ELF5 binding motifs [65]. This implies that the decrease in ELF5 expression with age in luminal epithelial cells results in the introduction of DNA methylation to the presumed binding sites [66]. Methylated DNA is believed to be the default state in the genome, rather than unmethylated DNA [67] because CpG sites are protected from methylation due to occupation by transcription factors. Furthermore, Oyer et al. reported that employing a tet-responsive promoter system to transiently suppress gene expression by adding doxycycline initially leads to the loss of active histone modification, followed by the methylation of the promoter region to stably suppress gene expression [68]. Taken together it suggests that the decline of *ELF5* age in luminal epithelial cells is not the consequence but rather the cause of aging. Thus, the decline of *ELF5* expression could be a key process to establish an aging, or accelerated aging, phenotype that may increase susceptibility to BC.

While *BRCA*1 and *BRCA2* are recognized as DNA repair proteins [69,70], it remains challenging to fully understand the mechanisms driving predominant breast and ovarian cancers associated with these mutations. Long-term treatment with progesterone and estrogen, rather than estrogen alone, in hormone replacement therapy is linked to increased risk of BC [71,72]. Tumor development in *BRCA1* knockout mice was reduced by inhibiting RANK/RANKL signaling using *Rank*:*Brca1*:*p53* triple-knockout mice [73], indicating that activation of the RANK/RANKL signal promotes tumor development. Progesterone plays a role in mammary gland differentiation through RankL-mediated induction of ELF5 in luminal progenitors [35]. Moreover, women carrying pathogenic *BRCA1* mutations exhibit higher progesterone levels compared to those at average risk [74], suggesting that continuous RANK/RANKL signaling due to elevated progesterone levels in *BRCA1* carriers could be a potential mechanism contributing to the development of breast cancer. Additionally, acquiring an accelerated aging phenotype in high-risk women that increases susceptibility to breast cancer could further promote cancer development in these women. The interplay between hormonal dynamics, particularly elevated progesterone levels in BRCA1 mutation carriers, the RANK/RANKL signaling pathway, and ELF5 may provide a crucial mechanistic link to the increased breast cancer risk observed in these individuals. The association of BRCA1/2 mutations with DNA repair deficiency, combined with hormonal influences on mammary gland differentiation and an accelerated aging phenotype, underscores the multifaceted nature of cancer development. Understanding these complex interactions is vital for developing targeted interventions that could mitigate the heightened susceptibility to breast cancer in high-risk populations.

In the transition to menopause significant changes in hormone levels occur, infamously an acute drop of estrogen at menopause [75,76], and the changes contribute to physical and biological consequences that may increase susceptibility to BC. One could assume that the aging hallmarks in mammary epithelia, including decline of *ELF5* expression and increased keratin 14 expression in luminal epithelia, is due to the changes in hormone levels. However, these and other hallmark aging changes in mammary epithelia occurs in women with pathogenic germline variants of *BRCA1*, *BRCA2*, or *PALB2* even in the young to middle-age range who have pre-menopause status [15,64]. Aging is an inevitable and gradual process, not an acute event. The change of breast tissue composition, including a decrease in epithelial cells and fibroconnective tissues, increased fat, and increased lobular involution (not pregnancy-associated involution), is observed starting in the fourth decade of life [77]. The median age of natural menopause among white women from industrialized countries ranges between 50 and 52 years [78]. Thus, the aging hallmarks we observed in older women at average risk and young to middle age genetically high-risk women were unlikely due to the changes of hormone levels at menopause. *ELF5* is downstream of progesterone signaling in mice [35], and the progesterone level in serum decreases earlier than menopause [66]. Therefore, age-associated *ELF5* downregulation may be correlated with progesterone levels. However, genetically high-risk women show on average higher levels of progesterone than average-risk women [74], and *ELF5* expression nevertheless decreases at a rate that is 10 to 40 times faster than average risk women [15]. Regulating ELF5 expression through progesterone may become more complex as women age, and not exactly modeled by mice, likely due to the gradual and persistent shifts in the cellular microenvironment that influence the phenotype of luminal epithelial progenitor cells [15,79].

Understanding the modulation of ELF5 expression within luminal epithelia and cKIT+ luminal progenitors is important for developing therapies targeted at age-related breast cancer prevention. scRNA-seq studies confirmed the presence of ELF5 in luminal epithelial cells and cKIT+ luminal progenitors in normal breast tissue. Yet, the specific patterns of ELF5 expression—whether it decreases uniformly, declines within certain subgroups, or varies by location in the breast tissue during aging—remain unclear. Clarifying how aging regulates ELF5-expressed cells will be instrumental in designing strategies to target ELF5, its downstream genes, and its regulatory pathways. Evidence suggests that both luminal epithelial cells and cKIT+ luminal progenitors increase in abundance with age [14,80], which implies that age-associated downregulation of ELF5 might occur across all luminal cells expressing ELF5, not just within a specific subgroup. One study suggests that the distribution pattern of cKIT+ luminal progenitors in the mammary gland tends to become more heterogeneous with age, a phenomenon particularly pronounced within lobular structures [81], underscoring the need to investigate how the distribution of ELF5-expressed cells changes in different breast regions—like large ducts, terminal ducts, and lobules—as aging progresses. In conclusion, a comprehensive understanding of ELF5 expression dynamics across the aging breast landscape is essential to inform targeted interventions for age-related breast cancer pathologies.

## 5. ELF5 as a Breast Specific Biomarker of Biological Age and Risk

Among women diagnosed with BC, a significant majority (90%) do not exhibit inherited mutations or have a family history of the disease, underscoring the predominance of sporadic over inherited BC cases [82,83]. Current genetic screening techniques are limited, identifying at-risk women based only on a small spectrum of monogenic risk factors. Mammography remains the most prevalent screening modality for BC, yet its sensitivity is approximately 70% [84]. This efficacy is further diminished in women with dense breast tissue [85]. Therefore, a comprehensive BC prevention strategy necessitates the development of risk assessment biomarkers tailored for early detection. These biomarkers should be capable of identifying and predicting subgroups particularly vulnerable to developing BC. Notably, elevated blood levels of IGF-1 and testosterone have been associated with increased BC risk [86,87,88]. Despite this correlation, blood-based risk prediction tools do not encapsulate the entire spectrum of BC risk factors. Oxidative stress may play a crucial role in carcinogenesis, including BC [89]. Studies have indicated an age-related increase in 8-oxo-2′-deoxyguanosine (8-oxo-dG), an oxidized derivative of deoxyguanosine [90,91]. Elevated levels of 8-oxo-dG in urinary excretion have been observed in BC patients [92], suggesting its potential as a biomarker for BC detection. Nevertheless, oxidative stress is a phenomenon occurring across tissues, posing a challenge to identifying tissue-specific diseases through non-invasive methods, such as urine samples [93,94]. Given the tissue-specific nature of cancer, our research pivots towards breast tissue biology-derived biomarkers that integrate the intersecting biologies of aging and genetic predisposition.

Epigenetic aging rates can be measured by DNA methylation profiles, as indicated by earlier studies [95,96]. Horvath’s pan-tissue clock, which estimates biological age using 353 CpG sites across various tissues, has been linked to age-related diseases [95]. However, its application to breast tissue has shown poor calibration and significant prediction errors. Notably, a considerable median absolute difference exists between estimated biological and chronological ages in breast tissue, possibly influenced by menstrual cycles among other factors.

*ELF5*’s expression decreases in luminal epithelia with age, coinciding with DNA methylation changes in its promoter [15,16]. Notably, *ELF5* expression is inversely proportional to promoter methylation with age. We have developed an *ELF5*-based biological clock that integrates these factors to predict breast-specific biological age accurately, with minimal errors (Figure 1A) [15]. This clock also independently identifies individuals at high risk of BC, independent of the specific monogenic risk factors. Remarkably, a significant reduction in *ELF5* expression was observed in 12.3% of the average-risk population, closely reflecting the BC incidence rate, which suggests the potential of the ELF5 clock to pinpoint high-risk individuals in a general population.

While specialized epigenetic clocks for breast tissue have been devised to predict accelerated biological aging in the breast [15,97,98,99], their development has relied on surgically removed breast tissues and primary cultured human mammary epithelial cells, which may not be ideal for clinical-scale screening. Blood or cell-free DNA might reveal specific DNA methylation changes, but it is uncertain whether these changes reflect biology that is relevant to breast tissue. The requirement for ample materials is a limitation of DNA methylation profiling, and while single-cell sequencing exists, it is impractical for clinical screening. Hence, the development and validation of the ELF5 clock, or other breast-specific clocks, will benefit from the use of clinically available samples, like random periareolar fine needle aspiration (RPFNA). We summarized the advantages and disadvantages associated with available human samples for implementing biological clocks that rely upon direct sampling of breast tissue (Figure 2). RPFNA, a minimally invasive procedure, is well-suited for assessing short-term breast cancer risk in asymptomatic women. Additionally, the RPFNA technique is employed for evaluating the effects of interventional studies of cancer prevention. These biopsies encompass a mixture of epithelial, stromal, adipose, and immune cells, making them suitable for gene expression analysis through qRT-PCR [100,101]. Leveraging RPFNA for the measurement of the ELF5 clock aligns with clinical workflows and also promises an accurate reflection of biology within the breast tissue milieu.

## 6. Challenges and Future Directions

The potential role of Omega-3 fatty acid, abundant in marine algae and fish, in reducing BC risk has been mentioned [103]. A high composition of eicosapentaenoic acid (EPA) and docosahexaenoic acid (DHA) in erythrocyte membrane has been demonstrated to be associated with reduced risk of breast cancer in Japanese cohorts [104]. Therefore, exploring the use of natural products such as Omega-3 or other natural products for potential breast cancer prevention is intriguing, given their negligible side effects and potential for other health benefits. The unique age and risk-associated biology of ELF5 in the breast suggests it may have distinct uses in breast cancer prevention and early detection. The primary goal of cancer prevention trials is to reduce cancer incidence, which can be difficult to measure due to the long timespans required to do so. Prevention trials also require large sample sizes to detect a difference in cancer incidence, as the baseline risk of developing cancer in a healthy population is relatively low [105]. This means a much larger and more diverse population is needed to demonstrate effectiveness, as compared to treatment trials which focus on patients with a specific stage or type of cancer. The *ELF5*-based biological clock could play a role in supporting breast cancer chemoprevention clinical trials by enabling more refined patient stratification. For example, identifying women whose breast tissue exhibits a biological age that is substantially accelerated compared to their chronological ages [15,64]. Furthermore, an indicator of biological activity from a prospective chemoprevention treatment could be a slowing of the rate of change of *ELF5* expression or promoter methylation, as measured in RPFNAs (Figure 1C). Because an acceleration of the ELF5 clock is a characteristic of some people at higher-than-average risk for BC, it could be used for early detection. Identifying individuals at higher risk based on their breast tissue’s biological age might allow for earlier interventions. The ELF5 clock could provide a nuanced and biologically relevant framework for conducting breast cancer prevention trials, addressing many of the challenges associated with these types of studies. However, the efficacy and practicality of implementing such a clock needs thorough validation in clinical settings.

## 7. Conclusions

ELF5 emerges as a potential breast-specific biomarker for age and risk, with its downregulation associated with an aging phenotype and heightened susceptibility to breast cancer. Moving forward, further research is warranted to elucidate the complex modulation of ELF5 expression in luminal epithelia and cKIT+ luminal progenitors during aging. Investigating ELF5-expressed cell distribution patterns across different breast regions and understanding the interplay between ELF5 and accelerated aging will be crucial. Additionally, refining the ELF5 clock as a breast-specific biological age predictor requires validation with clinically available samples, such as RPFNA, to ensure practicality and accuracy.

The potential applications of the ELF5 clock in breast cancer prevention trials hold promise for refining patient stratification, assessing biological activity during chemoprevention treatments, and enabling early detection. Thorough validation in clinical settings is essential for establishing the efficacy and feasibility of implementing the ELF5 clock as a valuable tool in breast cancer prevention strategies. Overall, understanding ELF5’s intricate role in breast biology and cancer susceptibility opens avenues for targeted interventions and personalized approaches in breast cancer prevention and early detection.

## Figures and Tables

**Figure 1 cancers-16-00431-f001:**
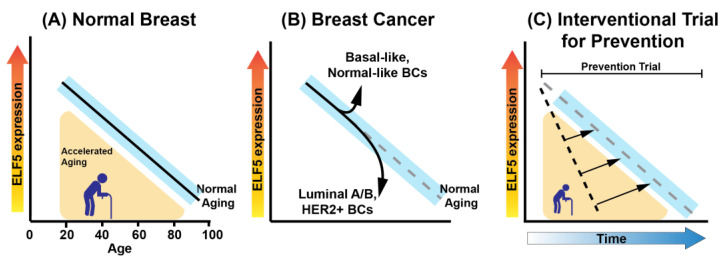
*ELF5* expression as a biomarker in breast tissue. These graphs depict models of *ELF5* expression states in mammary epithelia, based on findings from Miyano et al.’s Cancer Prevention Research 2021 [15]. Light blue depicts the confidence interval of a linear regression shown by the solid black line. (**A**) *ELF5* expression decreases with age in normal luminal epithelial cells. Women who carry certain monogenic risk factors for breast cancer exhibit lower levels of *ELF5* expression than would be predicted by their chronological age. Using *ELF5* expression levels as a biological clock, it may be possible to identify women who are high risk because they show accelerated biological aging in their breast tissue. Their expression levels would be in the yellow shaded area. (**B**) *ELF5* expression in different breast cancers subtypes appear in some ways as a caricature of aging. E.g. Basal-like breast cancers, which trend towards younger age at onset, have high *ELF5* expression. Whereas *ELF5* expression in luminal A or B breast cancer subtypes, which trend towards an older age at onset, show very low expression of *ELF5*. (**C**) Depicts the potential for utilization of *ELF5* expression as a clinical correlative in prevention interventional trials. Prospective forms of prevention that exert a desired biological effect would be predicted to slow, or even reverse, the rate of *ELF5* decrease over time.

**Figure 2 cancers-16-00431-f002:**
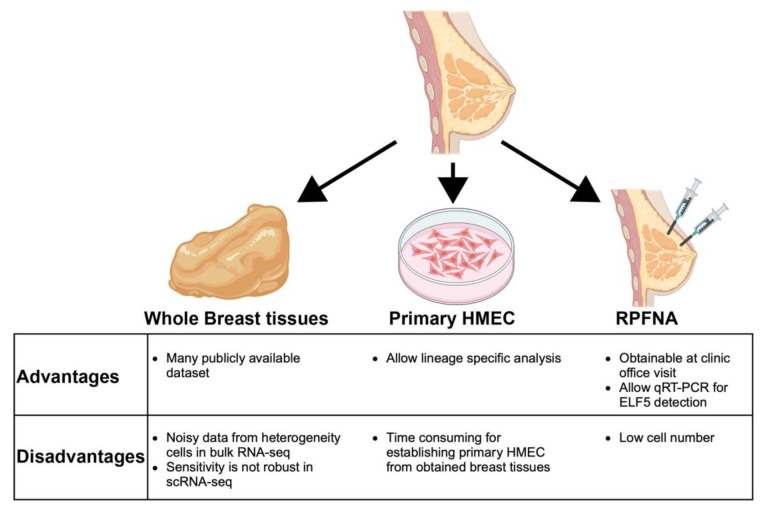
Tissue sources for *ELF5* measurements. We have measured *ELF5* expression in multiple sources of breast tissue, each with their associated advantages and disadvantages. HMEC; human mammary epithelial cells, RPFNA; random periareolar fine needle aspiration. Whole breast tissue is available as surgical discard material from prophylactic and cosmetic surgeries, which offers a plentiful source of normal tissue from across the human adult lifespan. There are a number of publicly available omics-datasets derived from whole breast tissue that enabled exploration of the *ELF5* clock, the major drawback being that lineage-specific signals (e.g., methylome or transcriptome) measured in whole breast tissue can be obscured by noise from associated lineages. Primary HMEC, when uncultured or cultured in low stress media [102], enable examination of individual lineages at high enrichment. Cultures can be time consuming and there are inevitable adaptations to the cell culture environment. RPFNAs can be obtained during office visits, enable longitudinal sampling, and there is sufficient cellular material to perform PCR and immunofluorescence assays. Drawbacks include discomfort, resulting sometimes in women declining the procedure, low cell number that limits the type and number of performable assays, and the proportions of adipose, blood, and epithelia are not controllable.

**Table 1 cancers-16-00431-t001:** Summary of *ELF5* expression and *ELF5* promoter DNA methylation in BC subtypes, along with pathways regulated by ELF5. TNBC; triple negative breast cancer, EMT; epithelial-mesenchymal transition, IFN-γ; Interferon-gamma.

Breast Cancer Type	*ELF5* Expression [15,25]	*ELF5* Promoter DNA Methylation [15]	Pathway
Basal/TNBC	High	Lower methylation	EMT suppressor [52,53]/promoter [54] IFN-γ signaling [53]
HER2	Low	Higher methylation	Unknown
Luminal A/B	Low	Higher methylation	EMT suppressor [52]/promoter [55]
Tamoxifen-resistantLuminal A	High	Unknown	Modulation of ERα-driven gene expression [56]

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
