# Peer review of "ELF5: A Molecular Clock for Breast Aging and Cancer Susceptibility"

_cancers, 2024, doi:10.3390/cancers16020431_

Round 1
Reviewer 1 Report
Comments and Suggestions for Authors
comments to the manuscript cancers-2814569, it is an interesting manuscript where the authors highlight the function of the ELF5 transcription factor in normal breast development and its involvement in aging and cancer. The mechanism of ELF5 expression is analyzed, and its potential as a biomarker for specific biological age of the breast and breast cancer. Some suggestions are described below.
Figure 1, enter the reference number corresponding to the citation, line 78.
Define ELF5 from the summary line 22
Homogenizer ELF5, Elf5 or italics
Figure 2 define Pros, cons, HMEC, RPFNA in the figure caption
Include a figure summarizing the interaction pathways of ELF5 in breast cancer.
Author Response
Reviewer 1
Comments and Suggestions for Authors
comments to the manuscript cancers-2814569, it is an interesting manuscript where the authors highlight the function of the ELF5 transcription factor in normal breast development and its involvement in aging and cancer. The mechanism of ELF5 expression is analyzed, and its potential as a biomarker for specific biological age of the breast and breast cancer. Some suggestions are described below.
Figure 1, enter the reference number corresponding to the citation, line 78.
Thank you we added the reference number to the Figure 1 legend.
Define ELF5 from the summary line 22
We included a brief explanation for ELF5 in Summary (line 21 to 24)
Homogenizer ELF5, Elf5 or italics
We adhere to the formatting guidelines provided by the Human Genome Organization and Mouse Genome Informatics for gene and protein names. Gene symbols are italicized, while protein symbols are non-italicized. In the case of humans, all gene symbols are presented in upper-case, whereas for rodents, only the initial letter is capitalized. We think this nomenclature serves to assist readers in distinguishing information derived from human or mouse subjects, aiding in the comprehension of gene and protein findings.
Figure 2 define Pros, cons, HMEC, RPFNA in the figure caption Include a figure summarizing the interaction pathways of ELF5 in breast cancer.
We revised Figure 2, incorporating an explanation for the abbreviations “HMEC” and “RPFNA” in the legend. Additionally, we added a new table (Table 1) summarizing ELF5, including pathways regulated by ELF5 in breast cancer subtypes.

Reviewer 2 Report
Comments and Suggestions for Authors
The authors in their article entitled: ELF5: A Molecular Clock for Breast Aging and Cancer Susceptibility have put their attention toward aging society and related female hormone dependent diseases (berets cancer - BC) in the light of potential diagnostic marker ELF5. Authors correctly perceive that even though obesity is the most spectacular disorder that requires immediate holistic action/solution the other related medical problems should be deeply investigated. One of the above is the increase in the numbers of BC which requires immediate and high-fidelity test development which has been presented clearly in the manuscript. Therefore, the authors in the correct way fill up the "orphan’s thesis" that early diagnosis is the key to therapy successful. It should be pointed out here that the references included in the article are correctly selected and support the authors' ideas. Breast cancer is one of the most common cancers related to aging (gender dependent). In their studies, authors presented a link between ELF5 level and breast cancer development as a function of time (aging). Therefore, the other molecular marker together with BRCA1/2 for BC investigated is highly desired for early diagnosis. The presented review article was well-written and readable. Moreover, the settled figures complement intuitively the scientific text making it understandable for common people. Due to the above, I hope that it will be a valuable position for a broad audience – as an open-access publication. Questions that should be answered: The authors should put some statistical information about BC detection not only in the USA but also in other countries. Additionally, some paragraphs on the diet protection role should be added. The level of 8-oxo-dG as a marker of DNA damage should be mentioned in the context of BC and compared with ELF5 level.
Author Response
Reviewer 2
The authors in their article entitled: ELF5: A Molecular Clock for Breast Aging and Cancer Susceptibility have put their attention toward aging society and related female hormone dependent diseases (berets cancer - BC) in the light of potential diagnostic marker ELF5. Authors correctly perceive that even though obesity is the most spectacular disorder that requires immediate holistic action/solution the other related medical problems should be deeply investigated. One of the above is the increase in the numbers of BC which requires immediate and high-fidelity test development which has been presented clearly in the manuscript. Therefore, the authors in the correct way fill up the "orphan’s thesis" that early diagnosis is the key to therapy successful. It should be pointed out here that the references included in the article are correctly selected and support the authors' ideas. Breast cancer is one of the most common cancers related to aging (gender dependent). In their studies, authors presented a link between ELF5 level and breast cancer development as a function of time (aging). Therefore, the other molecular marker together with BRCA1/2 for BC investigated is highly desired for early diagnosis. The presented review article was well-written and readable. Moreover, the settled figures complement intuitively the scientific text making it understandable for common people. Due to the above, I hope that it will be a valuable position for a broad audience – as an open-access publication. Questions that should be answered: The authors should put some statistical information about BC detection not only in the USA but also in other countries.
We have revised the manuscript and included sections discussing BC incidences in other countries/regions (line 30-34).
Additionally, some paragraphs on the diet protection role should be added.
We have revised the manuscript and included sections discussing diet protection role for BC (line 422-427).
The level of 8-oxo-dG as a marker of DNA damage should be mentioned in the context of BC and compared with ELF5 level.
Thank you for your suggestion. We have revised the manuscript and included sections discussing 8-oxo-dG as a potential marker for cancers (line 362-368). With the availability of multiple reliable biomarkers for BC risk assessment, there is potential to enhance prediction accuracy. Exploring the correlation between 8-oxo-dG levels and ELF5 expression could be an intriguing avenue. However, we acknowledge that this topic may seem somewhat peripheral in the context of this review manuscript. As a result, we briefly mention 8-oxo-dG as a BC biomarker without delving into a detailed comparison with ELF5 level.

Reviewer 3 Report
Comments and Suggestions for Authors
The manuscript primarily addresses how ELF5 expression changes with aging in breast tissue and its implications for breast cancer susceptibility. It explores the potential of ELF5 as a biomarker for indicating breast-specific biological age and cancer risk.
The topic is both original and highly relevant. It addresses a specific gap by focusing on the transcription factor ELF5, which has been less studied in the context of breast aging and cancer. The manuscript provides a significant understanding of breast tissue aging and its implications for cancer, an area of significant clinical importance.
The authors review the role of ELF5 in breast biology, aging, and cancer, combining the aspects of aging, genetic predisposition, and cancer susceptibility. They offer insights into potential prevention and early detection strategies.
The conclusions are consistent with the evidence and arguments presented. The authors effectively address the main question, highlighting the potential of ELF5 as a biomarker for breast aging and cancer susceptibility. However, they could emphasize the need for empirical validation of the ELF5 clock in clinical settings.
The references appear to be appropriate and relevant, covering a broad range of studies that support the manuscript's narrative. They are up-to-date and provide a solid foundation for the arguments made.
Overall, the manuscript is a comprehensive and insightful exploration of ELF5's role in breast biology, aging, and cancer. With some improvements, it could significantly impact the field.
Specific comments:
1) While the structure of the manuscript is logical, it could benefit from a clearer demarcation of sections to guide the reader through the narrative more smoothly.
For example: Introduction -> Breast Biology and Aging -> ELF5 in Normal Breast Development -> ELF5 in Breast Aging and Cancer -> ELF5 as a Biomarker -> Challenges and Future Directions -> Conclusion
2) Please include a more detailed explanation for Figure 2.
3) Where possible, please simplify the language to make the review more accessible to readers not specialized in the field.
Author Response
Reviewer 3
The manuscript primarily addresses how ELF5 expression changes with aging in breast tissue and its implications for breast cancer susceptibility. It explores the potential of ELF5 as a biomarker for indicating breast-specific biological age and cancer risk. The topic is both original and highly relevant. It addresses a specific gap by focusing on the transcription factor ELF5, which has been less studied in the context of breast aging and cancer. The manuscript provides a significant understanding of breast tissue aging and its implications for cancer, an area of significant clinical importance. The authors review the role of ELF5 in breast biology, aging, and cancer, combining the aspects of aging, genetic predisposition, and cancer susceptibility. They offer insights into potential prevention and early detection strategies. The conclusions are consistent with the evidence and arguments presented. The authors effectively address the main question, highlighting the potential of ELF5 as a biomarker for breast aging and cancer susceptibility. However, they could emphasize the need for empirical validation of the ELF5 clock in clinical settings. The references appear to be appropriate and relevant, covering a broad range of studies that support the manuscript's narrative. They are up-to-date and provide a solid foundation for the arguments made. Overall, the manuscript is a comprehensive and insightful exploration of ELF5's role in breast biology, aging, and cancer. With some improvements, it could significantly impact the field. Specific comments:
- While the structure of the manuscript is logical, it could benefit from a clearer demarcation of sections to guide the reader through the narrative more smoothly.
For example: Introduction -> Breast Biology and Aging -> ELF5 in Normal Breast Development -> ELF5 in Breast Aging and Cancer -> ELF5 as a Biomarker -> Challenges and Future Directions -> Conclusion
Thank you for your valuable comment. However, restructuring the entire manuscript is challenging during the reviewing process. Instead, we have focused on updating section titles to better align with the content.
- Please include a more detailed explanation for Figure 2.
We have updated the manuscript, incorporating a more comprehensive legend for Figure 2.
- Where possible, please simplify the language to make the review more accessible to readers not specialized in the field.
Thank you for your comment. We have simplified where possible.

Round 2
Reviewer 3 Report
Comments and Suggestions for Authors
The manuscript is now ready for publication.